# Availability of Physical Activity Tracking Data from Wearable Devices for Glaucoma Patients

Sonali B. Bhanvadia [1,2,†], Leo Meller [1,2,†], Kian Madjedi [3,4], Robert N. Weinreb [1] and Sally L. Baxter [1,2,*]

1 Division of Ophthalmology Informatics and Data Science, Viterbi Family Department of Ophthalmology and Shiley Eye Institute, University of California San Diego, La Jolla, CA 92093, USA; sbhanvad@health.ucsd.edu (S.B.B.); l5tang@health.ucsd.edu (L.M.); rweinreb@health.ucsd.edu (R.N.W.)
2 Health Department of Biomedical Informatics, University of California San Diego, La Jolla, CA 92093, USA
3 NIHR Biomedical Research Centre, Moorfields Eye Hospital NHS Foundation Trust and UCL Institute of Ophthalmology, London EC1V 2PD, UK; kian.madjedi1@ucalgary.ca
4 Department of Ophthalmology, University of Calgary, Calgary, AB T2N 1N4, Canada
* Correspondence: s1baxter@health.ucsd.edu
† These authors contributed equally to this work.

**Abstract:** Physical activity has been found to potentially modulate glaucoma risk, but the evidence remains inconclusive. The increasing use of wearable physical activity trackers may provide longitudinal and granular data suitable to address this issue, but little is known regarding the characteristics and availability of these data sources. We performed a scoping review and query of data sources on the availability of wearable physical activity data for glaucoma patients. Literature databases (PubMed and MEDLINE) were reviewed with search terms consisting of those related to physical activity trackers and those related to glaucoma, and we evaluated results at the intersection of these two groups. Biomedical databases were also reviewed, for which we completed database queries. We identified eight data sources containing physical activity tracking data for glaucoma, with two being large national databases (UK BioBank and *All of Us*) and six from individual journal articles providing participant-level information. The number of glaucoma patients with physical activity tracking data available, types of glaucoma-related data, fitness devices utilized, and diversity of participants varied across all sources. Overall, there were limited analyses of these data, suggesting the need for additional research to further investigate how physical activity may alter glaucoma risk.

**Keywords:** glaucoma; physical activity; fitness tracker; data availability; big data; wearables; exercise; vision; ophthalmology

## 1. Introduction

Glaucoma is a chronic optic neuropathy characterized by progressive retinal ganglion cell degeneration [1]. It is the leading cause of irreversible blindness globally, affecting nearly 70 million people and causing bilateral blindness in nearly 10% of those affected [1–3]. While intraocular pressure (IOP) is the primary target of treatment for glaucoma, IOP lowering alone often is not completely protective of glaucoma [4–7]. Furthermore, some patients develop glaucoma in the absence of high IOP. Hence, there is a need to understand the role of other modifiable lifestyle risk factors, including their potential impact on glaucoma pathophysiology, and also their potential as targets for therapeutic intervention.

Recently, an increasing number of studies have examined the potential association between exercise and glaucoma. There have been differing effects reported between aerobic and anaerobic exercise. In a short-term study with optical coherence tomography angiography (OCTA) by Nie et al., 25 eyes from patients were examined with primary open-angle glaucoma (POAG), and found that moderate aerobic exercise of 20 min (running) was sufficient to increase macular blood flow and lower the IOP in POAG patients [8]. Similar

results have been shown with more regular exercise. In a prospective interventional study by Siang et al., 45 healthy participants were enrolled in a 6-week aerobic exercise program, and their baseline IOP significantly decreased from a mean of 15.55 mmHg to 13.36 mmHg ($p < 0.001$), while the age- and gender-matched control cohorts who were not subject to the exercise program did not experience IOP reduction [9]. However, exercise has also been postulated to increase IOP, due to elevated blood pressure and increased aqueous humor production [9,10]. In a prospective study evaluating the association between IOP and weightlifting in the form of leg press, Vaghefi et al. reported transient IOP increases that returned to baseline within seconds after weight release across various loads, with an average increase of 26.4 mmHg during maximal muscular engagement [11]. Another study showed a similar trend for bench press exercises [12]. As individuals may often engage in both anaerobic and aerobic physical exercise, the relationship between overall physical activity and glaucoma remains mixed and needs to be elucidated through long-term surveillance of fitness activity.

This literature gap may be addressed through fitness surveillance by utilizing wearable activity trackers such as smartwatches, accelerometers, and fitness trackers. There has been an explosion in the utilization of wearable activity trackers, with an estimated 1444% increase in global shipping between 2014 and 2020, and nearly USD 2.8 billion spent worldwide on such devices in 2020 [13]. The data provided by these wearable activity trackers may empower long-term, robust analyses of large cohorts on the effects of exercise on glaucoma, which can strengthen the existing evidence on this topic that has largely been built upon small cohorts. However, little is known regarding the availability and characteristics of such databases for research utilization. In this study, we performed the first scoping review and query of data sources providing wearable physical activity data for glaucoma patients, as well as a synthesis of existing research on this topic, in order to provide insights on the potential for large-scale analyses of exercise and glaucoma, and directions for future research.

## 2. Methods

We conducted a scoping review of available physical activity tracking data for patients with glaucoma in literature databases, which included PubMed and MEDLINE using the National Library of Medicine, and population-based biomedical databases. The population-based biomedical database search was conducted across various online engines, including Google, Safari, and Google Scholar.

### 2.1. Search Strategy

The overall framework for the search strategy involved examining the intersection between two concept sets: one related to physical activity tracking (concept set 1), and one related to glaucoma (concept set 2) (Figure 1). The search terms for concept set 1 encompassed those related to physical activity tracking, including *smartwatch*, *physical activity tracker*, *wearable*, and *fitness tracker*. The following brand names of physical activity trackers were also included in this concept set: *FitBit*, *Apple watch*, *Garmin*, *ActiGraph*, *Amazfit*, *BioIntelliSense*, *Biofourmis*, *BodiMetrics*, *BodyMedia*, *CardiaInsight*, *Empatica*, *Garmin*, *Huawei*, *iRhythm*, *Motiv*, *Omron*, *Oura*, *Oxiline*, *Philips*, *Rockley*, *Sensomics*, *Wellue*, *Withings*, *Xiaomi*, *Verily*, and *VitalConnect*. The search terms for concept set 2 encompassed those related to glaucoma, including *glaucoma*, *glaucoma suspect*, *ocular hypertension*, *primary open-angle glaucoma (POAG)*, and *elevated intraocular pressure (IOP)*. For impartiality, two reviewers (S.B.; L.M) independently examined the identified sources with terms that fell into either concept set 1 or concept set 2 to identify sources that reported both glaucoma and physical activity tracking data, and extracted information relevant for each dataset.

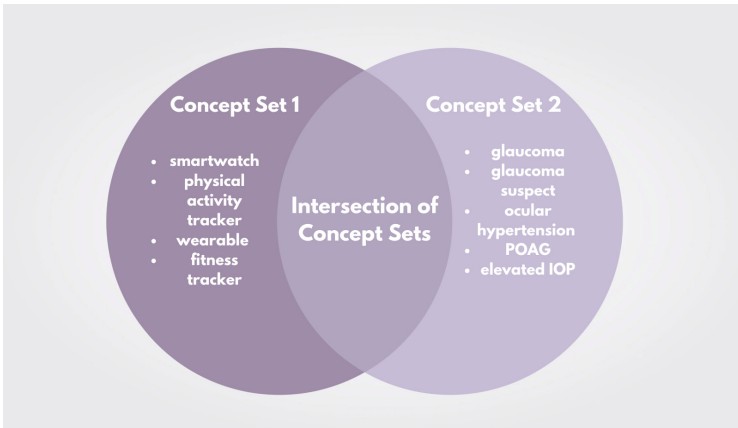

**Figure 1. Search strategy utilized in scoping review.** Search terms of concept set 1 and concept set 2 are reported; the intersection of both represents the datasets of interest. POAG: primary open-angle glaucoma. IOP: intraocular pressure. Interventionary studies involving animals or humans, and other studies that require ethical approval must list the authority that provided approval and the corresponding ethical approval code.

## 2.2. Eligibility Criteria

In order to be considered for review, we only included journal articles and databases with datasets pertaining to both physical activity tracking and participants with a glaucoma-related diagnosis. We were primarily interested in physical activity tracking data encompassing multi-day follow up and not just observation periods with short durations (i.e., ~30 min of exercise). For the literature search, only original research, case studies, case series, prospective and retrospective studies, and clinical trials were included. Review articles, those not in English, articles unavailable to the study team, letters to the editor, and clinical trial proposals were excluded. For the database search, we only included population-based biomedical databases. For databases that required registration, we completed registration and access requirements where applicable. All of the data sources had to be accessible to either the public or by registration, or upon request to authors of individual journal articles.

## 2.3. Data Query and Extraction

Two reviewers independently conducted data queries of the final included data sources from both the literature search and database search for impartiality (Figure 2). We recorded the characteristics of each database, including their access details, type of available glaucoma data, race/ethnicity of participants in the overall dataset (if available), and the fitness device(s) used. We also queried data to identify the number of patients with glaucoma, the number of patients with fitness tracking data available (any diagnosis), as well as the intersection of both.

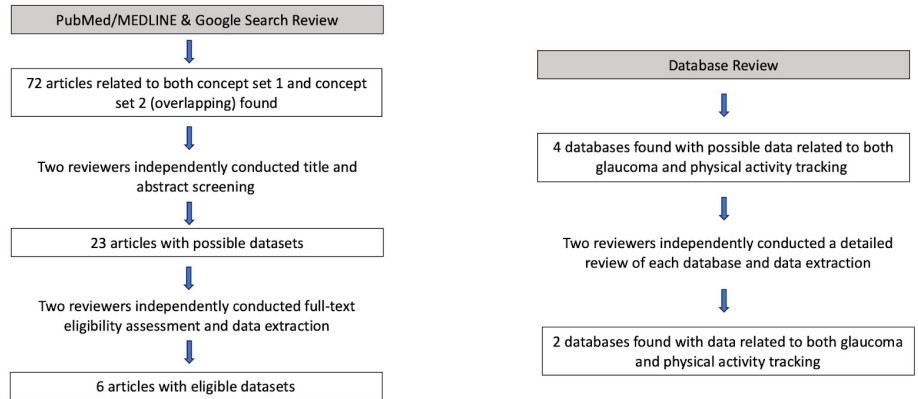

**Figure 2. Flowchart for article and database review.** A total of 72 articles were screened and 6 articles were included, while 4 databases were examined and 2 were included.

## 3. Results

### 3.1. Overview of Available Datasets

There was a total of eight datasets that reported physical activity tracking data for patients with a glaucoma diagnosis or related condition (Figure 2). These datasets included both peer-reviewed journal articles (6/8) identified through literature search with data available upon request, as well as large population-based biomedical datasets (2/8) including the National Institutes of Health *All of Us* research program and the UKBB [14,15]. The detailed characteristics of all eight data sources are described in Table 1. Access to both large datasets was restricted by varying requirements, including registration, citizenship, established research profiles, training, and payment.

### 3.2. Cohort Sizes

Among the datasets, the UKBB had the largest number of participants with physical activity tracking data, with 37,641 participants, followed by the *All of Us* research program with 12,844. *All of Us* had the largest number of participants with a glaucoma diagnosis, with 15,578. Of the 15,578 participants with glaucoma in the *All of Us* dataset, 2.8% (437) had physical activity tracking data available. Of the 7550 participants with a glaucoma diagnosis in the UKBB dataset, 5.7% (429) had physical activity tracking data available.

Ma et al. measured the change in ocular perfusion pressure and intraocular pressure after short and long-term exercise with the use of a sports watch [16]. They reported that physical activity reduced intraocular pressure and increased ocular perfusion pressure in this study. E et al. used accelerometers to track the physical activity of their participants, and reported that participants who were encouraged to participate in physical activity at home and felt confident in performing the physical activity showed significant benefits to the IOP and OPP, especially those with a severe glaucoma diagnosis [17]. Ramulu et al., 2012 and Lee et al. both used the same datasets [18,19]. Ramulu et al., 2012 measured the physical activity performance amongst those with and without glaucoma, and although they found no significant differences, participants with reduced physical activity and numbers of steps had a greater visual field loss [18]. Lee et al. concluded that the participants with greater visual field loss showed lower performance for physical activity, which could be a risk factor for glaucoma [19]. In the article by Berry et al., participant data were extracted from the UKBB and from the PROGESSA study group [20]. The authors in this study concluded that based on participant data available from the PROGRESSA database, physical activity tracking showed that physical activity was associated with slower macular GCIPL thinning, showing benefits of exercise for eye conditions.

### 3.3. Main Findings from Journal Articles

Regarding the large databases, there were no journal articles using data from the *All of Us* Research Program, and there were two journal articles analyzing the UKBB. Berry et al. focused on the association between physical activity and rates of macular thinning; they examined data from 8862 eyes from the UKBB and reported significant positive associations between physical activity (data obtained through accelerometer) and the cross-sectional total macular thickness in POAG patients (beta = 0.8 μm/standard deviation) [20]. On the other hand, Madjedi et al. from the Modifiable Risk Factors for Glaucoma Collaboration and the UKBB Eye and Vision Consortium examined the association between physical activity (data also obtained through accelerometer), glaucoma, and its related characteristics [21]. Contrary to expectations, the authors reported no association between time spent in physical activity or physical activity level and glaucoma status, no association between high levels of physical activity and IOP, and no association with macular retinal nerve fiber layer thickness, yet more time in moderate and vigorous physical activity was associated with a thicker macular ganglion cell-inner plexiform layer [21].

Among the smaller cohort studies, the prospective analysis by E et al. provided the highest number of participant data among the identified journal articles; they tracked the

physical activity and activity locations of 229 patients from the Falls in Glaucoma Study with suspected glaucoma or glaucoma for 7 days through accelerometers and global positioning systems [17]. The authors highlighted the need of providing an activity-friendly non-home environment for glaucoma patients, and the importance of a safe home environment [17]. Ramulu et al. tracked the physical activity of 83 glaucoma patients and 53 control patients for 7 days and found no significant difference in physical activity between the glaucoma and control patients [18]. The authors later followed up with data from this cohort l; they found higher physical activities such as walking, moderate–vigorous physical activity, and non-sedentary activity were associated with slower rates of VF loss in a glaucoma population [19]. Finally, the randomized clinical trial conducted by Ma et al. provides long-term exercise data from 252 eyes of 123 POAG patients [16]. The POAG patients were randomized into exercise (n = 61) and control groups (n = 62), with the exercise group jogging 30 min in the morning between 6–10 a.m. with at least 20 jogging sessions per month for 3 months. Overall, the authors reported a decreasing trend for IOP for the exercise group, where after 3 months of exercise their 24 h IOP level was mostly lower than baseline.

### 3.4. Ophthalmological Data Available among Datasets with Physical Activity Tracking

The ophthalmological information that was available differed within each dataset. The UKBB had the most abundant ophthalmological data available, which includes visual acuity testing, autorefraction and keratometry, IOP, corneal hysteresis, corneal resistance factor, corneal biomechanical property-adjusted IOP data from 117,649 participants, as well as retinal photography and spectral domain optical coherence tomography (SD-OCT) data from 68,151 participants. The *All of Us* database contained more epidemiological level data, including glaucoma status, family history, genetics, diagnosis codes, and procedure codes. Currently, there are no data related to IOP, visual fields, or other imaging available on the *All of Us* Researcher Workbench. The information from large databases are supplemented by unique data provided by individual journal articles, such as cup–disc ratio and ocular perfusion pressure.

### 3.5. Features of Physical Activity Tracking Data

The most common fitness devices utilized for physical activity tracking were Fitbit and other unspecified accelerometers. The Fitbit data were used in the *All of Us* and UKBB datasets, which included steps and heart rate. The accelerometer used by Berry et al. counted steps for physical activity tracking, which was similar to Ramulu et al. and Lee et al., which also used an accelerometer to monitor the average number of steps per day, along with the duration of physical activity of differing [18–20]. Accelerometers were also used to determine the active minutes and steps per day for participants in the study described by E et al.; they also used global positioning system trackers to determine the location (at-home vs. away from home), although it is not specified if these trackers are built into or separate from the accelerometers [17]. Ma et al. used an omnidirectional accelerometer to track physical activity via steps and heart rate of participants during periods of time where physical activity was being performed [16].

### 3.6. Diversity of Data

The *All of Us* dataset had the most diverse population of participants in the overall database, with 244,540 (52.0%) of their database composed of participants that identify as White, and 48.0% being combined minority groups, including 82,020 (17.4%) Black, 75,420 (16%) Hispanic, and 14,800 (3.1%) Asian. On the other hand, the UKBB database primarily consisted of White subjects (511,441, 94.4%), and only 5.6% of combined racial minorities, including 10,253 (1.89%) Asian, and 8290 (1.53%) Black. The participant diversity among each individual journal article varied and had limited granularity, with Ma et al. reporting homogenous Chinese participants [16].

**Table 1.** Data sources with physical activity tracking data for individuals with glaucoma.

| Dataset | Number of Glaucoma Patients | Number of Patients (Any Diagnosis) with Physical Activity Tracking Data Available | Number of Glaucoma Patients with Physical Activity Tracking Data Available | Access Details | Glaucoma-Related Data Available in Dataset | Fitness Device Used | Overall Race/Ethnicity Composition of Dataset |
|---|---|---|---|---|---|---|---|
| National Institutes of Health *All of Us* Research Program [22] | 15,578 | 12,844 | 437 | Restricted access by registration and approval, must be a U.S. resident, training required, requires institutional data use agreement | Epidemiological analysis including glaucoma status, family history, and genetics. Diagnosis codes and procedure codes are also available | FitBit | 244,540 (52.0%) White, 82,020 (17.4%) Black/African American/African, 75,420 (16%) Hispanic/Latino, 14,800 (3.1%) Asian, and Other (11.5%) |
| United Kingdom BioBank (UKBB [a]) [23] | 7550 * | 37,641 | 429 | Restricted by registration and requires payment, requires affiliation with research/university, must have an established research profile | Visual acuity, autorefraction and keratometry, IOP [b], corneal hysteresis, corneal resistance factor, retinal photography, spectral domain optical coherence tomography | Accelerometer data (various devices possible) | 511,441 (94.4%) White, 10,253 (1.89%) Asian/British Asian, 8290 (1.53%) Black/British Black, Other (2.13%) |
| Ma QY et al. (2022) [16] | 123 | 61 | 61 | Journal article, data available upon request to the authors | IOP [b], ocular perfusion pressure | Unspecified sports watch | 123 (100%) Asian (Chinese) |
| E et al. (2021) [17] | 229 | 229 | 229 | Journal article, data available upon request to the authors | Visual acuity, visual field damage, integrated visual field sensitivity | Waist-bound physical tracking device with GPS tracking | 63 (27.5%) African American |
| Ramulu et al. (2012) [18] | 83 | 141 | 83 | Journal article, data available upon request to the authors | Better-eye visual field mean deviation | Omnidirectional accelerometer | 40 (28.4%) African American |

**Table 1.** *Cont.*

| Dataset | Number of Glaucoma Patients | Number of Patients (Any Diagnosis) with Physical Activity Tracking Data Available | Number of Glaucoma Patients with Physical Activity Tracking Data Available | Access Details | Glaucoma-Related Data Available in Dataset | Fitness Device Used | Overall Race/Ethnicity Composition of Dataset |
|---|---|---|---|---|---|---|---|
| Lee et al. (2019) [19] | 83 | 141 | 83 | Journal article, data available upon request to the authors | Eye Mean deviation, visual field loss | Omnidirectional accelerometer | 46 (32%) Non-Caucasian |
| Berry et al. (2023) [20] | 512 (PROGRESSA [c]) N/A (UK BB) | 465 (PROGRESSA) 96,679 (UK BB) | 465 (PROGRESSA) N/A (UK BB) | Journal article, data accessible via UKBB [a] and available upon request to the PROGRESSA study group | Visual acuity, IOP, ultrasound central corneal pachymetry, vertical cup to disc ratio (PROGRESSA) Described above (UKBB) | FitBit (PROGRESSA) Triaxial accelerometer (UKBB) | Not reported by authors, available upon request to the PROGRESSA study group and under UKBB. |
| Madjedi et al. (2023) [21] | Refer to UKBB [a] data above | Refer to UKBB [a] data above | Refer to UKBB [a] data above | Journal article, data accessible via UKBB [a] | Refer to UKBB [a] data above | Refer to UKBB [a] data above | Refer to UKBB [a] data above |

[a] UKBB: UK BioBank. [b] IOP: intraocular pressure. [c] PROGRESSA: progression risk of glaucoma: relevant SNPs with significant association. * This number will differ based on the way in which glaucoma is defined by various researchers and is not used ubiquitously across all UKBB research. For this study, this number is accurate.

### 4. Discussion

There is increasing interest in understanding the role of lifestyle factors in glaucoma, specifically the role of exercise or physical activity. Overall, we reported on eight data sources, two of which are large national databases and six are from individual journal articles. The United States National Institutes of Health *All of Us* Research Program had the greatest number of glaucoma patients with physical activity tracking data available, followed by the UKBB. Individual journal articles also provided physical activity data for individuals with glaucoma.

The UKBB is a large, open-access, continuously updated, population-based biomedical database providing health and genetic data from 500,000 UK participants. Specifically, the UKBB Eye and Vision Consortium reports rich, comprehensive data related to eye and vision in the UKBB. However, despite their rich data on eye and vision and physical activity tracking data, only two existing articles within the literature (both published in 2023) utilized the UKBB to consider the role of exercise with glaucoma (Berry et al. and Madjedi et al., findings described above) [20,21]. Moreover, only Madjedi et al. focused on the association between exercise and glaucoma status, and reported overall that there was no association between physical activity and glaucoma and IOP. This large population-based study by Madjedi et al. represents one of the first attempts to utilize wearable physical activity data to address the association between exercise and glaucoma, and provides contradictory results to previous smaller-scale studies [8,9,21]. Therefore, not only do the limited studies conducted thus far in the UKBB on physical activity and glaucoma call for future investigations to utilize this robust database, this opposition of results by Madjedi et al. with previous literature also calls for the need for future studies to further assess the association between exercise and glaucoma utilizing wearable physical activity tracking data.

Although the *All of Us* research database contained the highest number of glaucoma patients with physical activity tracking data available, currently, no analysis on physical activity and glaucoma has been conducted utilizing this robust database. Contrary to the UKBB, where Caucasian participants comprise more than 94% of the available overall data points, the *All of Us* database is significantly more diverse [23]. Specifically, of the overall cohort enrolled in *All of Us*, 17.4% of the participants identify as Black, 16.0% as Hispanic, over 50% participants from the *All of Us* database are racial and ethnic minorities, and more than 80% are underrepresented in biomedical research [22]. With this diverse participant population, the availability of glaucoma diagnosis and procedure information, and wearable fitness tracking data available, there are opportunities for better understanding the associations between exercise and glaucoma using data from the *All of Us* Research Program. However, a key challenge for this data source is the lack of ocular phenotypic information such as visual field information and imaging, although there are ongoing conversations regarding adding ocular imaging to future versions of the dataset [24].

The data contained within individual journal articles also provide limited information about the relationship between exercise and glaucoma. Although E et al. did not focus on the overall association between physical activity and glaucoma, considering their relatively robust sample size and ample vision evaluation data including VF damage, this dataset provides invaluable information for future analysis [17]. In addition, Ramulu et al., 2012 and Lee et al. of the same participant pool suggest potential for further investigation and collaboration in addressing the complex relationship between exercise and glaucoma [18,19]. Finally, the randomized clinical trial conducted by Ma et al. contains a relatively robust sample size (252 eyes of 123 POAG patients), provides long-term exercise data, and contains IOP and ocular perfusion pressure data [16]. Thus, the dataset contained within this clinical trial represents a potential opportunity to conduct a pooled analysis with the other available data sources we identified in this study. Given the increasing attention on studying modifiable lifestyle factors that modulate glaucoma risk, the clinical significance of the present study is that we provide a comprehensive review and database query to update clinicians on the latest evidence on the association between physical activity and glaucoma, and highlight potential databases that can be used for further examination

of this relationship. Our results may better inform clinicians on prescribing exercise to glaucoma patients.

Thus, future investigations on physical activity and glaucoma should not only aim to validate our existing understanding of their overall association, but should also provide a more granular analysis based on the characteristics of the exercise, which may have a different impact on glaucoma risk. While many existing studies on this topic focused on aerobic interventions (i.e., treadmill) of short durations, exercise in reality can be complex and take varying forms [8,25,26]. Although it is generally believed that acute aerobic exercise reduces intraocular pressure, aerobic exercise such as swimming may in fact increase IOP, mediated by the usage of swimming goggles [26–28]. In addition, yoga, an Increasingly popular exercise that may be classified as aerobic or anaerobic depending on the intensity, may lead to transient increases in IOP for 2 min, especially during head-down positions, and it is unknown whether this may increase glaucoma risk [29]. Moreover, as discussed previously, classical anaerobic exercise, such as weightlifting, has been associated with an increase in IOP [11,12]. Therefore, the existing understanding of the association between physical activity and glaucoma has been mostly confined to the investigation of one particular type of exercise, with limited understanding of how different intensities, durations, types of exercise, and combinations of these factors may modulate glaucoma risk. This knowledge gap can be addressed through analysis of the data from wearable fitness activity trackers, as many smartwatches allow participants to select the activity type, and can thus offer granular data based on the type, intensity, and duration of exercise. If any significant association between a particular exercise type and glaucoma risk is found, smart device notifications may even serve as a platform for potential risk modification. Overall, future research on physical activity and glaucoma must focus on delineating the differential impact of various exercises and their associated characteristics, and the emergence of smart, wearable fitness activity trackers serves a critical role in addressing this goal. Finally, another gap identified in the review was the lack of specification regarding the type of accelerometer used for data collection. While the validity of many commercially available accelerometers has been established with prior studies, there can still be variations in measurement [30–33]. Thus, future research on this topic should report the specific model type and validity of the fitness tracker utilized to ensure reproducibility and accuracy of the results. In addition, the final included articles did not maximize the full potential of physical activity tracking data, which may have been dependent on the methods and software used to analyze the dataset. Specifically, physical activity tracking data may be utilized to derive parametric model of the circadian clock (both amplitude and phase) and non-parametric 24-scale domains of activity (most active 10-hour period (M10); least active 5-hour period (L5); onset time; inter-daily stability, intra-daily variability, relative amplitude) [34]. This is particularly important; given that glaucoma patients may present with altered circadian rhythms, the timing of their physical exercise may prove to be different [35]. As the characteristics of physical exercise (duration, timing including onset, offset, phase) are associated with outdoor light exposure, this is expected to be affected in POAG patients, given their retinal ganglion cell degeneration; this can subsequently alter measures derived from physical activity tracking devices and circadian regulation [36,37]. Hence, in future research, particular attention should also be made to physical activity, tracking wearable devices that provide data on light and blue light exposure [38]. In order to best provide evidence-based recommendations to clinicians on physical activity and glaucoma, future studies may consider the following: (1) Recruiting matched exercise and control group participants, and separately examining any associations in both the healthy and glaucoma populations; (2) considering the impact of other environmental conditions, such as light exposure, and circadian rhythms; (3) following the participants in a longitudinal manner, for at least 32 weeks, and prescribing different types of physical activity (subtypes of aerobic and anaerobic), tracking the changes in glaucoma-related characteristics, most notably IOP, and following up with the participants to re-examine any associations between physical activity and glaucoma using electronic health care record

diagnosis codes; and (4) validating the findings under participants from different and diverse demographics.

Our study has some limitations. Firstly, given the access restrictions associated with each database and the data availability of each journal article, we were unable to provide a more detailed description of the full range of data associated with the data sources. Secondly, some qualified studies and databases may not have been included despite our rigorous search with detailed key terms, and the included studies and data sources may prove heterogeneous in the type and level of data they offer for future pooled analyses. Thirdly, we were only able to offer a cross-sectional snapshot of the current data availability on physical fitness data and glaucoma, while the landscape of data sources may be continuously updated due to the rapid emergence and popularity of wearable physical activity tracking devices. For example, the *All of Us* Research Program has ongoing enrollment, and has rapidly increased in cohort size over the last few years; cohort sizes will continue to grow in the coming years. Despite these limitations, this first scoping review and data query of physical activity tracking data available in glaucoma patients identified important gaps to conduct further analyses, and synthesized existing literature to suggest future directions and trends for research.

## 5. Conclusions

We identified multiple available datasets, including two large publicly available biomedical databases that offer opportunities for analyses regarding the relationship between physical activity and glaucoma. However, limited analyses of these datasets exist in the current literature. Future analyses should consider leveraging these data sources and focus on delineating the differential impact of various exercise types on glaucoma risk. Physical activity tracking among healthy and glaucoma patients could provide beneficial information on the association between exercise and glaucoma, as well as lead to future interventions related to physical activity for patients at high risk for glaucoma.

**Author Contributions:** Conceptualization, S.B.B., L.M. and S.L.B.; methodology, S.B.B., L.M. and S.L.B.; validation, K.M., R.N.W. and S.L.B.; formal analysis, S.B.B., L.M. and K.M.; investigation, S.B.B., L.M. and K.M.; resources, S.B.B., L.M. and K.M.; writing—original draft preparation, S.B.B. and L.M.; writing—review and editing, S.B.B., L.M., K.M., R.N.W. and S.L.B.; supervision, R.N.W. and S.L.B.; project administration, R.N.W. and S.L.B.; funding acquisition, R.N.W. and S.L.B. All authors have read and agreed to the published version of the manuscript.

**Funding:** This study was supported by National Institutes of Health (NIH) grants T35 EY033704, DP5OD029610, P30EY022589, R01EY034146, R01MD014850, and an unrestricted departmental grant from Research to Prevent Blindness.

**Data Availability Statement:** Data accessed in this study are available from their corresponding sources, either upon registration or request.

**Acknowledgments:** The authors wish to thank investigators from the Artificial Intelligence Ready and Equitable Atlas for Diabetes Insights project (AI-READI, ai-readi.org) for information regarding common brands of physical activity tracking devices that contributed to the search strategy. The *All of Us* research program is supported (or funded) by grants through the National Institutes of Health, Office of the Director: Regional Medical Centers: 1 OT2 OD026549; 1 OT2 OD026554; 1 OT2 OD026557; 1 OT2 OD026556; 1 OT2 OD026550; 1 OT2 OD 026552; 1 OT2 OD026553; 1 OT2 OD026548; 1 OT2 OD026551; 1 OT2 OD026555; IAA #: AOD 16037; Federally Qualified Health Centers: HHSN 263201600085U; Data and Research Center: 5 U2C OD023196; Biobank: 1 U24 OD023121; The Participant Center: U24 OD023176; Participant Technology Systems Center: 1 U24 OD023163; Communications and Engagement: 3 OT2 OD023205; 3 OT2 OD023206; and Community Partners: 1 OT2 OD025277; 3 OT2 OD025315; 1 OT2 OD025337; 1 OT2 OD025276. In addition to the funded partners, the *All of Us* research program would not be possible without the contributions made by its participants. This research has been conducted using data from UK Biobank, a major biomedical database.

**Conflicts of Interest:** The authors declare no conflict of interest.

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
