# Peer review of "Availability of Physical Activity Tracking Data from Wearable Devices for Glaucoma Patients"

_information, doi:10.3390/info14090493_

Round 1
Reviewer 1 Report
This paper provides overview of the availability of physical activity tracking data from wearable devices for glaucoma patients. The authors came to the conclusion that still there are limited analyses of such data, and additional research to further investigate how physical activity may alter glaucoma risk is needed.
The authors may wish to consider the following comments to enhance some important aspects of tracking activity in glaucoma patients in discussion.
-
Information, that can be retrieved from the data, collected by wearable devices depends mainly on the software tools applied to analyze the data. The referenced papers did not use fully potential of the data, e.g. in-depth examination of parametric circadian (amplitude, phase) and non-parametric 24-scale domains (maximum 10-hour, M10; least 5-hours. L5; M10, L5onset time; Inter-daily variability, intra-daily stability, relative amplitude) of activity that can be derived from activity data (doi: 10.1080/07420528.2017.1337032). Glaucoma patients have compromised circadian rhythms, and that is likely that timing of physical activity can be altered (doi.org/10.3390/app12189220; doi: 10.3389/fneur.2020.584479). Also, duration of physical activity and particularly its timing (phase, onset, offset) are closely related to outdoor light exposure that is expected to be affected in POAG. In patients with ganglion cells loss or compromised retinal ganglion cell function certain actimetry derived measures are affected that are not just amount of physical activity per se (doi: 10.3109/07420528.2012.691146; doi: 10.1038/s41598-020-58205-1; doi: 10.1097/IJG.0000000000002186).
-
Since light exposure and light perception are of primary interest in glaucoma patients, and light and activity are closely related aspects of circadian physiology, it should be noted that future research should pay special attention to data from the wearables that provide data on light / blue light exposure (doi.org/10.3390/app122211794).
-
P.4 3.2. Main Findings from Journal Articles. Discussing confronting results from papers by Berry et al,m 2023 (ref.20) and Majedi et al., 2023 (ref. 21), it can be noted that in a paper by Majedi while no association between physical activity and macular retinal nerve fiber layer thickness was found, greater time spent in moderate and vigorous physical activity still associated with macular ganglion cell–inner plexiform layer thickness.
Author Response
Point 1: Information, that can be retrieved from the data, collected by wearable devices depends mainly on the software tools applied to analyze the data. The referenced papers did not use fully potential of the data, e.g. in-depth examination of parametric circadian (amplitude, phase) and non-parametric 24-scale domains (maximum 10-hour, M10; least 5-hours. L5; M10, L5onset time; Inter-daily variability, intra-daily stability, relative amplitude) of activity that can be derived from activity data (doi: 10.1080/07420528.2017.1337032). Glaucoma patients have compromised circadian rhythms, and that is likely that timing of physical activity can be altered (doi.org/10.3390/app12189220; doi: 10.3389/fneur.2020.584479). Also, duration of physical activity and particularly its timing (phase, onset, offset) are closely related to outdoor light exposure that is expected to be affected in POAG. In patients with ganglion cells loss or compromised retinal ganglion cell function certain actimetry derived measures are affected that are not just amount of physical activity per se (doi: 10.3109/07420528.2012.691146; doi: 10.1038/s41598-020-58205-1; doi: 10.1097/IJG.0000000000002186).
Response 1: Thank you very much for these insightful suggestions. We have taken in consideration of these recommendations and references, which have now been included in the discussion (Paragraph 6). We discussed how included articles do not maximize the full potential of physical activity tracking, and how this data can be used to derive a model of the circadian clock, since circadian rhythms in glaucoma patients can be altered (and can affect physical activity and circadian regulation).
Point 2: Since light exposure and light perception are of primary interest in glaucoma patients, and light and activity are closely related aspects of circadian physiology, it should be noted that future research should pay special attention to data from the wearables that provide data on light / blue light exposure (doi.org/10.3390/app122211794).
Response 2: Thank you for this suggestion. We also discussed this in the discussion (Paragraph 6) and have taken into consideration the references/suggestions mentioned, such as how circadian rhythms in glaucoma patients can be altered due to light exposure/perception, and the importance for future research to be conducted on wearable devices that provide data on light and blue light exposure.
Point 3: P.4 3.2. Main Findings from Journal Articles. Discussing confronting results from papers by Berry et al,m 2023 (ref.20) and Majedi et al., 2023 (ref. 21), it can be noted that in a paper by Majedi while no association between physical activity and macular retinal nerve fiber layer thickness was found, greater time spent in moderate and vigorous physical activity still associated with macular ganglion cell–inner plexiform layer thickness.
Response 3: Thank you for this comment. In the results section (Now 3.4), we have addressed this suggestion in regards to the paper by Madjedi et al., and have explained how more time in moderate and vigorous physical activity was associated with thicker macular ganglion cell-inner plexiform layer, even though no association with macular retinal nerve fiber layer thickness was found
Reviewer 2 Report
The manuscript entitled Availability of physical activity tracking data from wearable devices for glaucoma patients is a scoping review of available physical activity tracking data for patients with glaucoma in literature databases. The theme of this manuscript is interesting; it seems important as future ideas for studies. For clinical practice and practitioners, it is not so useful due to the important limitations of this search. Probably, a systematic review with these datasets will be more interesting for clinical practice.
However, this is the first paper, which has used data from the All of Us Research Program.
The manuscript seems to be a narrative review; it is well written. The method are well explain.
On page 7 there is a figure which is probably the flowchart of this review. Please put the legend and insert this figure in the text.
Author Response
Point 1: On page 7 there is a figure which is probably the flowchart of this review. Please put the legend and insert this figure in the text.
Response 1: Thank you for this suggestion. We have added in a legend to the figure (Figure 2) and inserted a reference to the figure into the results (3.2.).
Reviewer 3 Report
Thanks for allowing me to review this paper. In this study, the authors conducted a scoping review to find wearable physical activity data availability for glaucoma patients. This is an exciting study but needs much improvement to be considered for publication. Here are my comments:
1. Why have the authors conducted a literature search only in PubMed? Why not other databases like Scopus and Web of Science?
2. Need to rearrange the result section. Please start with search findings (Figure and table), then describe what you have got from the literature.
3. Please describe the clinical implications of this study in the discussion part.
Minor editing is needed.
Author Response
Point 1: Why have the authors conducted a literature search only in PubMed? Why not other databases like Scopus and Web of Science?
Response 1: Thank you for this comment. Given the limited articles on this topic and the nature of the present narrative review focused on biomedical and life sciences literature, we found the PubMed search to be comprehensive. In fact, we did supplement our PubMed search with MEDLINE with a google search as well to improve the scope of our review.
Point 2: Need to rearrange the result section. Please start with search findings (Figure and table), then describe what you have got from the literature.
Response 2: Thank you for this suggestion. We have rearranged the results so that the table and figure are at the start of the results section followed by discussion of the literature flow.
Point 3: Please describe the clinical implications of this study in the discussion part.
Response 3: Thank you for this insightful comment. We have addressed the clinical implications of this study in the end of the fourth paragraph of the discussion: Given the increasing attention on studying modifiable lifestyle factors that modulate glaucoma risk, the clinical significance of the present study is that we provide a comprehensive review and database query to update clinicians on the latest evidence on the association between physical activity and glaucoma and highlight potential databases that be used for further examination of this relationship. Our results may better inform clinicians on prescribing exercise to glaucoma patients.
Reviewer 4 Report
Availability of physical activity tracking data from wearable 2 devices for glaucoma patients is a novel original paper dealing with the effects of physical activity on glaucoma based on measurable data from activity tracking devices. The research is well-designed and conducted. Before publication only minor issues should be addressed and that is how a further studies should be designed to get evidence-based recommendation of physical activity effect on glaucoma. Which parameters should be measured? For how long? How the study should be set up if regarding all data available from previous studies no valid conclusions are drawn
Author Response
Point 1: Before publication only minor issues should be addressed and that is how a further studies should be designed to get evidence-based recommendation of physical activity effect on glaucoma. Which parameters should be measured? For how long? How the study should be set up if regarding all data available from previous studies no valid conclusions are drawn
Response 1: Thank you for this comment. We have addressed this in the discussion section (Paragraph 6). Specifically, we discussed how future studies should be conducted to generate evidence-based recommendations on this important topic, as well as the parameters that should be measured and study duration.
Round 2
Reviewer 3 Report
Thanks for the revised version. It can be considered for publication.
N/A
Author Response
Thank you for your feedback.